# Utility of Gallium-68-DOTATATE PET CT in Surveillance of Resected Gastroenteropancreatic NET

**DOI:** 10.3390/jcm14238545

**Published:** 2025-12-02

**Authors:** Kirstie Lithgow, Sunil Samnani, Caitlin T. Yeo, Denise Chan

**Affiliations:** 1Department of Medicine, University of Calgary, Calgary, AB T2N 1N4, Canada; 2Arnie Charbonneau Cancer Institute, University of Calgary, Calgary, AB T2N 1N4, Canada; 3Department Surgery, University of Calgary, Calgary, AB T2N 1N4, Canada; 4Department of Radiology, University of Calgary, Calgary, AB T2N 1N4, Canada

**Keywords:** neuroendocrine tumours, surgical resection, surveillance, radiation exposure, ^68^Ga-DOTATATE, PET/CT

## Abstract

**Background/Objectives:** For completely resected well differentiated (WD) gastroenteropancreatic (GEP) NET, guidelines differ in recommendations for utilization of SSTR-based functional imaging in post-operative surveillance. While ^111^In-Octreotide has previously been the standard of care, imaging with ^68^Ga-labelled peptides has expanded in recent years due to increased sensitivity to detect smaller volume diseases and reduced costs. Though many centres have widely adopted imaging with ^68^Ga-labelled peptides, its role in surveillance of resected GEP NET has not been well defined. We sought to characterize current utilization of imaging with ^68^Ga-DOTATATE PET CT (^68^Ga-DOTA) for post-operative surveillance of WD GEP NET and assess the impact on clinical management. **Methods**: We conducted a retrospective review of all ^68^Ga-DOTA scans performed from April 2019 to August 2024. Inclusion criteria were age ≥ 18 years with WD grade 1 and 2 GEP NET that had undergone curative-intent surgery, had Stage I-III disease at diagnosis, and had ^68^Ga-DOTA post-operatively. **Results**: Forty-six scans met the inclusion criteria. We identified four indications for ^68^Ga-DOTA: (1) post-operative assessment (n = 12); (2) routine surveillance (n = 18); (3) recurrence suspected based on cross-sectional imaging (n = 10); and (4) recurrence suspected based on biochemical monitoring (n = 6). Avidity for each indication was observed in 45%, 8%, 50%, and 80%, respectively. Initiation of long-acting somatostatin analogue was the most common management following avidity. **Conclusions**: ^68^Ga-DOTA best informed clinical decision making when there was clinical suspicion for residual or metastatic disease post-operatively or based on cross-sectional imaging or biochemistry. The utility of this modality for routine surveillance appears limited.

## 1. Introduction

Neuroendocrine tumours (NET) are neoplasms which can secrete peptides and neuroamines [1] and are most commonly found in the gastrointestinal system [2]. Current imaging tools used in evaluation of NET include anatomical imaging (ultrasound, CT, and MRI) and functional imaging such as ^111^In-Octreotide or ^68^Ga (Gallium)-labelled peptides [3]. ^111^In-Octreotide has historically been the standard of care for functional imaging in diagnosis and follow-up of NET, however drawbacks include long imaging time and decreased detection of lesions compared to CT and MRI [4]. Recently, there has been introduction of new somatostatin agonistic DOTA peptides such as DOTATATE that exhibit high affinity for somatostatin receptors and can be labelled with the positron emitting radioisotope ^68^Ga. Advantages to using ^68^Ga-labelled somatostatin agonist peptides over ^111^In-Octreotide include shorter imaging time, improved spatial resolution, a higher tumour to background uptake, reduced costs [3], and increased sensitivity for detecting smaller volume diseases compared to ^111^In-Octreotide [5,6,7,8,9,10,11].

For cases of completely resected WD (well differentiated) gastroenteropancreatic (GEP) NET, guidelines differ in recommendations for utilization of somatostatin receptor (SSTR)-based functional imaging in post-operative surveillance [12,13,14,15,16]. Recommendations for functional imaging are heterogenous, including at baseline only, after 13 to 36 months, every 24 months, or not specified [12,13,14,15,16]. Though many tertiary centres (including ours) have widely adopted functional imaging with ^68^Ga-labelled peptides in recent years, the role of this imaging in surveillance of completely resected WD GEP NET has not been well defined. We sought to characterize current utilization of imaging with ^68^Ga-DOTATATE PET CT (hereafter abbreviated as ^68^Ga-DOTA) for post-operative surveillance of resected WD GEP NET at our centre and assess the impact on clinical management.

## 2. Materials and Methods

Each patient received an IV injection of ^68^Ga-DOTA. Imaging was conducted 45–90 min after injection of 100 to 250 MBq ^68^Ga-DOTA. Images were collected for 10–15 min in a single bed position from skull base to mid-thigh using a PET/CT scanner. After acquisition was completed, CT scan was performed for attenuation correction and localization [3]. Images were interpreted by board physicians certified in both Nuclear Medicine and Diagnostic Radiology. SUVs were calculated for abnormal sites using software provided by the equipment manufacturer. Avidity is based on SUV for the DOTATATE PET/CT and Krenning score system. PET CT scanner is a GE Discovery MI with a 5 ring system, software version 3.01_1.SPO2, and 128 slice capabilities.

We conducted a retrospective study of all ^68^Ga-DOTA scans performed since this imaging modality became available at our centre in April 2019, inclusive until August 2024. All patients who undergo ^68^Ga-DOTA give informed consent and are entered into an institutional registry. Inclusion criteria were age ≥ 18 years with WD grade 1 and 2 GEP NET that had undergone curative-intent surgery, had Stage I-III disease at diagnosis, and had ^68^Ga-DOTA performed post-operatively. Records were individually reviewed to gather demographic data and determine indication for ^68^Ga-DOTA and subsequent clinical management.

This study was approved by the Health Research Ethics Board at the University of Calgary (HREBA.CC-22-0259, 13 January 2023; HREBA.CC-22-0259_REN1, 9 January 2024; HREBA.CC-22-0259_REN2, 10 December 2024). All participants that underwent imaging with ^68^Ga-DOTA gave informed consent to participate in the retrospective study. Descriptive statistics were performed for categorical (percentages) and continuous (median and IQR) variables.

## 3. Results

A total of 283 scans were performed during the study period with 46 meeting inclusion criteria. Exclusion reasons were stage IV disease (n = 128), pre-operative imaging (n = 62), and non-GEP primary (n = 47) and neuroendocrine carcinoma (n = 1). Baseline characteristics are shown in Table 1. A proportion of 58% (n = 26) had stage III disease, 11% (n = 5) stage II, 2% (n = 1) stage I, and the rest (n = 13) were unstaged due to unknown status of regional lymph nodes. Median follow-up was 51 months (IQR: 40) and median months from surgery to ^68^Ga-DOTA was 14 (IQR: 35).

We identified four indications for ^68^Ga-DOTA: (1) Post-operative assessment: defined as any scan performed within 12 months of surgery (n = 13); (2) Routine surveillance: defined as surveillance imaging performed >12 months post-operatively without a specific indication (n = 17); (3) Recurrence suspected based on cross-sectional imaging findings suspicious for recurrent or metastatic disease (n = 10); (4) Recurrence suspected based on biochemical monitoring with serum chromogranin A and/or 24 h urine 5HIAA (n = 6). Findings and impact on clinical decision making are summarized in Table 2. Clinical management included ongoing surveillance, surgical resection of residual or metastatic disease, or initiation of long-acting somatostatin analogue (SSA).

In studies performed for indication #1 (post-operative baseline) avidity was detected in 4/13 (28%) cases and suggestive of residual disease in four patients. Management consisted of initiation of medical therapy with long-acting somatostatin analogue (SSA) in two cases, repeat surgery in one case, and ongoing surveillance in one case. Staging was known for nine of these cases and notably, 6/9 (67%) had stage III disease.

In studies performed for indication #2 (routine surveillance), avidity was detected in one case (6%) which showed a small DOTATATE avid lesion with no anatomical correlate on cross-sectional imaging. No further evidence of metastatic disease has been detected during 42 months of subsequent follow-up; therefore, this is considered a false positive or clinically insignificant result.

In the studies performed for indication #3 (recurrence suspected based on cross-sectional imaging), avidity was detected in 7/10 (70%). Four confirmed avidity of liver lesions, leading to initiation of SSA (n = 3) or surgical resection (n = 1). Two confirmed avidity of mesenteric lymphadenopathy (one of these also had an additional bone lesion not identified on cross-sectional imaging) leading to initiation of SSA in the case the bone avidity and surveillance in the other. One case confirmed avidity of peritoneal metastases, leading to repeat surgery.

In studies performed for indication #4 (recurrence suspected based on biochemistry), avidity was detected in 5/6 (83%). In four of these, mesenteric lymphadenopathy or peritoneal deposits were detected leading to initiation of SSA (n = 3) or ongoing surveillance (n = 1). In the other case, avidity was related to hydrosalpinx which was deemed a false positive result. This scan showed avidity in the right lower posterior pelvis which does not have an anatomic correlate on prior CT and initially raised suspicion for peritoneal metastases. However, subsequent MRI showed no evidence of metastatic disease and clarified that the area of DOTATATE uptake corresponded to the hydrosalpinx which had been stable for several years.

Of the scans that demonstrated avidity, there were seven (15%) with findings clinically in keeping with residual or metastatic disease which did not have a correlate on cross-sectional imaging with MRI or CT including mesenteric lymphadenopathy (n = 4), pancreatic lesion (n = 1), bone lesion (n = 1), and peritoneal deposit (n = 1). Six of these were managed with initiation of SSA (n = 3 lymphadenopathy, n = 1 pancreatic, n = 1 peritoneal deposit, n = 1 bone) or surveillance (n = 1 lymphadenopathy).

## 4. Discussion

Guidelines differ in recommendations for functional imaging in surveillance of resected GEP NET [12,13,14,15,16,17]. Furthermore, most of these were authored prior to widespread adoption of imaging with ^68^Ga-labelled peptides, which have demonstrated improved detection over ^111^In-Octreotide in the detection of small-volume disease [5,6,7,8,9,10,11], and bone metastases [7,8,9]. The role of ^68^Ga-DOTA in surveillance of resected GEP NET requires further delineation, especially considering cumulative radiation exposure associated with surveillance imaging schedules [18]. To our knowledge, this is the first study to evaluate the clinical utility of ^68^Ga-DOTA for surveillance of completely resected GEP NET in a real-world context.

We examined clinical practice at our centre regarding the use of ^68^Ga-DOTA in post-operative surveillance of curative-intent resected GEP NET. At our centre, we do not have a defined protocol for utilization of ^68^Ga-DOTA; this is based on the discretion and clinical judgement of the ordering physician. Furthermore, this study includes the time frame during which ^68^Ga-DOTA was introduced at our centre and as such, physicians’ comfort level with using this modality and clinical practices may have evolved during the study interval. These factors likely contributed to the observed variation in practice patterns for utilization of this imaging modality. We identified four indications for performing this imaging in our practice setting. Overall, avidity was observed in 17 scans (37%), of which two were considered to be false positive results. Of the 15 scans that were clinically in keeping with residual or metastatic disease, these cases were managed by initiation of SSA (n = 9), surgery (n = 3), or surveillance (n = 3).

Although routine surveillance was the most common indication, avidity was demonstrated in only one case which was deemed a false positive result. In the setting of cancer surveillance, such results have potential to generate significant anxiety for patients and caregivers, as well as lead to downstream burden of additional investigations or unnecessary/inappropriate therapies. Conversely, usage of this imaging modality to correlate with suspicious biochemistry or findings on cross-sectional imaging was higher yield, often confirming metastatic disease and leading to change in clinical management.

Indication #2 was the second most common indication and avidity was detected in 31% of cases. This detection rate presumably reflects a high degree of clinical suspicion for residual disease on the part of the ordering provider, as ^68^Ga-DOTA is not routinely part of baseline post-operative imaging for completely resected GEP NET at our centre. Accordingly, most staged cases had confirmed stage III disease at the time of resection.

Though majority of avid lesions (85%) had correlates on cross-sectional imaging with CT or MRI, a number (n = 7) of lesions were detected on ^68^Ga-DOTA alone, in keeping with reports from previous studies [5,7,19]. However, the benefit of earlier detection of these lesions is unclear, given the indolent or slowly progressive nature of low-grade GEP NET.

In general, our findings lend further support to existing consensus recommendations. The most recent consensus guidelines from the Society of Nuclear Medicine & Molecular Imaging state that indications #3 and #4 are appropriate scenarios for utilization of SSTR-based imaging, as these can allow for selection of appropriate therapy and lend insight into tumour grade [17]. These guidelines state that indication #1 “may be appropriate”, with many suggesting that a single SSTR-based study could be considered after curative-intent resection, particularly if this was not performed pre-operatively. However, the authors highlight that there was a lack of consensus amongst the committee due to the potential for overuse of SSTR-based imaging, and the low clinical impact of detecting small-volume residual disease [17]. Finally, these guidelines highlight equipoise around the role of SSTR-based imaging for surveillance (“may be appropriate”) but suggested this should generally not be used for routine imaging or in place of cross-sectional imaging [17]. Our findings reinforce the low utility and potential pitfalls (false positive results) of SSTR-based imaging in a surveillance setting.

In addition to clinical indications for utilization of ^68^Ga-DOTA and other SSTR-based imaging, potential for radiation exposure and cost-effectiveness should also be considered. Previous work from our institution has highlighted that the cumulative radiation burden for patients undergoing imaging surveillance for resected GEP NET is significant and in the range associated with secondary malignancies [18]. The additional healthcare costs and burden to healthcare systems related to the use of ^68^Ga-DOTA should also be considered, particularly in settings where clinical management is unlikely to be impacted. While previous authors have established the cost-effectiveness of ^68^Ga-DOTA in diagnostic workup of NET [20], the cost-effectiveness of this imaging has not (to our knowledge) been established in the post-operative/surveillance setting [20].

The major strength of our study is we assessed real-world ordering practices and clinical decision making at a high-volume referral centre for GEP NET, increasing the applicability of our findings. However, there are several limitations. This is a small, single-centre experience with retrospective analysis. We did not have a dedicated surveillance imaging schedule or protocol, therefore ordering practices likely differ amongst providers. Our centre adopted the Gallium scan in 2019 and since then utilization of this modality has expanded and evolved. Our results may not be generalizable to imaging using other ^68^Ga-DOTA-peptides, to NET with non-GEP primaries, or to grade 3 NET. We did not collect or analyze quantitative data including SUVmax values and lesion based analyses, further limiting reproducibility. As such, this should be considered a pilot study which may not be broadly generalizable; findings should be interpreted with caution.

## 5. Conclusions

Our results suggest that ^68^Ga-DOTA best informed clinical decision making when there was clinical suspicion for residual or metastatic disease post-operatively or based on cross-sectional imaging or biochemistry. The utility of this modality as part of routine surveillance schedules for resected GEP NET appears limited and should be weighed against healthcare resources utilization, radiation exposure, and potential for false positive results.

## Figures and Tables

**Table 1 jcm-14-08545-t001:** Baseline characteristics of included cases.

	n = 64	
Sex	56% F (n = 25)	
Median age (IQR)	52 (22)	
Grade	1	67% (30)
	2	33% (16)
Primary		
• Small bowel	n = 25 (54%)	
• Pancreas	n = 8 (17%)	
• Appendix	n = 6 (13%)	
• Gastric	n = 3 (7%)	
• Liver	n = 2 (4%)	
• Colon/rectum	n = 2 (4%)	
Stage		
• I	n = 1 (2%)	
• II	n = 5 (11%)	
• III	n = 26 (57%)	
• Unknown	n = 14 (30%)	

**Table 2 jcm-14-08545-t002:** ^68^Ga-DOTA results and impact on clinical management (see text for details).

Indication	1. Post-Operative Baseline	2. Routine Surveillance	3. Suspected Metastatic Disease on Cross-Sectional Imaging	4. Suspected Metastatic Disease on Biochemistry
Scans	n = 13 (28%)	n = 17 (37%)	n = 10 (22%)	n = 6 (13%)
Timing of scan post-surgery (months): median (IQR)	3 (2)	17 (25)	33 (43)	66 (59)
Avidity	n = 4	n = 1	n = 7	n = 5
Clinical impact	• SSA n = 2 • Surgery n = 1 • Surveillance = 1	False positive result	Liver n = 4 • SSA n = 3 • Surgery n = 1 Lymphadenopathy n = 2 • Surveillance n = 1 • SSA n = 1 Peritoneal metastases n = 1 • Surgery n = 1	Lymphadenopathy n = 3 • SSA n = 2 • Surveillance n = 1 Peritoneal metastases n = 1 • SSA n = 1 False positive n = 1

## Data Availability

Data is unavailable due to privacy or ethical restrictions; however, an anonymized data set could be made available upon reasonable request.

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
