# Peer review of "Utility of Gallium-68-DOTATATE PET CT in Surveillance of Resected Gastroenteropancreatic NET"

_jcm, 2025, doi:10.3390/jcm14238545_

Round 1
Reviewer 1 Report
Comments and Suggestions for Authors
The main goal of this work was to analyse the current utilization of 68Ga-DOTATATE PET-CT imaging for post-operative monitoring of resected well differentiated gastroenteropancreatic neuroendocrine tumours and assess its impact on clinical management. The work is topical and could serve as general guidelines to clinics with limited/no experience with 68Ga-DOTA. However, given the small cohort analysed in this work, the results must be interpreted with caution. Further comments below:
- Please spell out all acronyms used in the Abstract (first time use). The same applies to the acronyms used for the first time in text (mostly in the Introduction).
- Materials and methods –some relevant information are missing from this section: the size of the cohort included in this study; the manufacturing details of the PET/CT device + software used.
- The flowchart in Fig 1 starts with a total nr. of 46 patients/scans, while the first sentence of Results states that 45 scans met the inclusion criteria. Please rectify the numbers so they match.
- Line 156 – replace ’preforming’ with ‘performing’
- Among limitations, the authors must mention the small cohort.
- Given the above (small cohort, further divided into 4 smaller groups) this work should be considered a pilot study with fairly low generalizability power, thus the conclusions require careful interpretation. This aspect should be mentioned in the Discussion / limitations.
Author Response
The main goal of this work was to analyse the current utilization of 68Ga-DOTATATE PET-CT imaging for post-operative monitoring of resected well differentiated gastroenteropancreatic neuroendocrine tumours and assess its impact on clinical management. The work is topical and could serve as general guidelines to clinics with limited/no experience with 68Ga-DOTA. However, given the small cohort analysed in this work, the results must be interpreted with caution. Further comments below:
- Please spell out all acronyms used in the Abstract (first time use). The same applies to the acronyms used for the first time in text (mostly in the Introduction).
Thank you, this has been done.
- Materials and methods –some relevant information are missing from this section: the size of the cohort included in this study; the manufacturing details of the PET/CT device + software used.
The cohort size is 283 (line 99). Other details have been added (lines)
- The flowchart in Fig 1 starts with a total nr. of 46 patients/scans, while the first sentence of Results states that 45 scans met the inclusion criteria. Please rectify the numbers so they match.
Thank you, 46 is the correct number and this has been clarified.
- Line 156 – replace ’preforming’ with ‘performing’
This has been done
- Among limitations, the authors must mention the small cohort.
This has been done
- Given the above (small cohort, further divided into 4 smaller groups) this work should be considered a pilot study with fairly low generalizability power, thus the conclusions require careful interpretation. This aspect should be mentioned in the Discussion / limitations.
Thank you, this has been added to the final paragraph of the discussion.
Reviewer 2 Report
Comments and Suggestions for Authors
The authors presented a paper about “Utility of Gallium-68-DOTATATE PET CT in surveillance of resected gastroenteropancreatic NET”.
The topic itself is interesting from a clinical point of view
1) Page 2 line lines 38-44: please remove the following unnecessary paragraphs
- How to Use This Template 38
The template details the sections that can be used in a manuscript. Note that each section has a corresponding style, which can be found in the “Styles” menu of Word. Sections that are not mandatory are listed as such. The section titles given are for articles. Review papers and other article types have a more flexible structure.
Remove this paragraph and start section numbering with 1. For any questions, please contact the editorial office of the journal or support@mdpi.com.
2) Imaging was performed at varying times post-surgery without a defined follow-up schedule, which weakens internal validity: please better specify the clinical reasons for such variations
3) Quantitative data such as SUVmax values and lesion-based analyses are missing, reducing reproducibility. The discussion should also integrate recent guideline updates and justify clinical thresholds for surveillance use.
4) The authors should provide more context on the implications of false positives and how they might affect patient outcomes.
5) The discussion highlights the utility of 68Ga-DOTA in specific clinical scenarios but could delve deeper into the cost-effectiveness and potential risks of radiation exposure.
Author Response
The authors presented a paper about “Utility of Gallium-68-DOTATATE PET CT in surveillance of resected gastroenteropancreatic NET”.
The topic itself is interesting from a clinical point of view
1) Page 2 line lines 38-44: please remove the following unnecessary paragraphs
- How to Use This Template 38
The template details the sections that can be used in a manuscript. Note that each section has a corresponding style, which can be found in the “Styles” menu of Word. Sections that are not mandatory are listed as such. The section titles given are for articles. Review papers and other article types have a more flexible structure.
Remove this paragraph and start section numbering with 1. For any questions, please contact the editorial office of the journal or support@mdpi.com.
Thank you, this has been done.
2) Imaging was performed at varying times post-surgery without a defined follow-up schedule, which weakens internal validity: please better specify the clinical reasons for such variations
We have expanded on this (lines 165 to 171)
3) Quantitative data such as SUVmax values and lesion-based analyses are missing, reducing reproducibility. The discussion should also integrate recent guideline updates and justify clinical thresholds for surveillance use.
This has been added as a limitation
We have integrated recent guideline updates (197 to 210)
4) The authors should provide more context on the implications of false positives and how they might affect patient outcomes.
We have expanded on this as suggested (183 to 186).
5) The discussion highlights the utility of 68Ga-DOTA in specific clinical scenarios but could delve deeper into the cost-effectiveness and potential risks of radiation exposure.
This has been done (lines 211 to 220).
Reviewer 3 Report
Comments and Suggestions for Authors
The manuscript “Utility of Gallium-68-DOTATATE PET/CT in Surveillance of Resected Gastroenteropancreatic Neuroendocrine Tumors” is clear, well-structured, and clinically meaningful. It addresses an important and timely question about the role of 68Ga-DOTATATE PET/CT in the postoperative surveillance of well-differentiated GEP-NETs—an area where current evidence is still limited and existing guidelines remain heterogeneous. The study provides valuable real-world data from a tertiary referral center, and the conclusions are, overall, well supported by the results presented.
To further strengthen the manuscript, I suggest a few revisions aimed at improving clarity and completeness:
Methods
Please clarify how eligible cases were identified (for example, through a PET scheduling database, institutional registry, or radiology archive).
Define “avidity” explicitly—was this based on visual assessment, a quantitative SUV threshold, or a consensus between readers?
It would also be helpful to indicate the typical follow-up period after each type of scan indication.
Results
Consider adding a concise summary table showing the indication for scanning, number of studies performed, rate of avid findings, and subsequent management outcomes. This would make the results section easier to follow at a glance.
Indicate whether any patients with initially negative scans later developed recurrence during follow-up.
Discussion
You might expand briefly on the implications of radiation exposure and cost/resource utilization, referring to Iannuzzi et al., Cancers 2024, which is already cited.
When mentioning that most avid lesions had correlates on cross-sectional imaging, specifying the exact proportion would make this statement more precise.
Minor issues
Correct a few typographical details (e.g., “its’ role” → “its role”).
Ensure consistent radionuclide formatting.
Briefly clarify the hydrosalpinx false-positive case for completeness, as this will help readers interpret the findings appropriately.
Author Response
The manuscript “Utility of Gallium-68-DOTATATE PET/CT in Surveillance of Resected Gastroenteropancreatic Neuroendocrine Tumors” is clear, well-structured, and clinically meaningful. It addresses an important and timely question about the role of 68Ga-DOTATATE PET/CT in the postoperative surveillance of well-differentiated GEP-NETs—an area where current evidence is still limited and existing guidelines remain heterogeneous. The study provides valuable real-world data from a tertiary referral center, and the conclusions are, overall, well supported by the results presented. To further strengthen the manuscript, I suggest a few revisions aimed at improving clarity and completeness: Methods Please clarify how eligible cases were identified (for example, through a PET scheduling database, institutional registry, or radiology archive).
This has been added (lines 87-88)
Define “avidity” explicitly—was this based on visual assessment, a quantitative SUV threshold, or a consensus between readers? It would also be helpful to indicate the typical follow-up period after each type of scan indication.
Thank you, this has been added to the new Table 2 Results
Consider adding a concise summary table showing the indication for scanning, number of studies performed, rate of avid findings, and subsequent management outcomes. This would make the results section easier to follow at a glance.
Thank you, we have changed Figure 1 to a summary table (Table 2) for better readability.
Indicate whether any patients with initially negative scans later developed recurrence during follow-up. Unfortunately, our database does not include this data, as it is am imaging database that only captured clinical data at the time of presentation and at the time of functional imaging. As such, we are unable to retrieve this data in a reasonable time frame.
Discussion You might expand briefly on the implications of radiation exposure and cost/resource utilization, referring to Iannuzzi et al., Cancers 2024, which is already cited.
This has been added (lines 212 to 221).
When mentioning that most avid lesions had correlates on cross-sectional imaging, specifying the exact proportion would make this statement more precise.
This has been clarified.
Minor issues Correct a few typographical details (e.g., “its’ role” → “its role”).
This has been done
Ensure consistent radionuclide formatting. Briefly clarify the hydrosalpinx false-positive case for completeness, as this will help readers interpret the findings appropriately.
This has been done.
Round 2
Reviewer 2 Report
Comments and Suggestions for Authors
The revised version has improved the quality of the article.